# The ability to classify patients based on gene-expression data varies by algorithm and performance metric

**Stephen R. Piccolo** *, **Avery Mecham**, **Nathan P. Golightly**, **Jérémie L. Johnson**, **Dustin B. Miller**

Department of Biology, Brigham Young University, Provo, Utah, United States of America

* stephen_piccolo@byu.edu

**Data Availability Statement:** Source code for each algorithm used can be found in repositories for the respective software libraries used in this study: * https://github.com/scikit-learn/scikit-learn * https://

## Abstract

By classifying patients into subgroups, clinicians can provide more effective care than using a uniform approach for all patients. Such subgroups might include patients with a particular disease subtype, patients with a good (or poor) prognosis, or patients most (or least) likely to respond to a particular therapy. Transcriptomic measurements reflect the downstream effects of genomic and epigenomic variations. However, high-throughput technologies generate thousands of measurements per patient, and complex dependencies exist among genes, so it may be infeasible to classify patients using traditional statistical models. Machine-learning classification algorithms can help with this problem. However, hundreds of classification algorithms exist—and most support diverse hyperparameters—so it is difficult for researchers to know which are optimal for gene-expression biomarkers. We performed a benchmark comparison, applying 52 classification algorithms to 50 gene-expression datasets (143 class variables). We evaluated algorithms that represent diverse machine-learning methodologies and have been implemented in general-purpose, open-source, machine-learning libraries. When available, we combined clinical predictors with gene-expression data. Additionally, we evaluated the effects of performing hyperparameter optimization and feature selection using nested cross validation. Kernel- and ensemble-based algorithms consistently outperformed other types of classification algorithms; however, even the top-performing algorithms performed poorly in some cases. Hyperparameter optimization and feature selection typically improved predictive performance, and univariate feature-selection algorithms typically outperformed more sophisticated methods. Together, our findings illustrate that algorithm performance varies considerably when other factors are held constant and thus that algorithm selection is a critical step in biomarker studies.

## Author summary

When a patient is treated in a medical setting, a clinician may extract a tissue sample and use transcriptome-profiling technologies to quantify the extent to which thousands of genes are expressed in the sample. These measurements reflect biological activity that may

github.com/mlr-org/mlr * https://github.com/
Waikato/weka-3.8 * https://github.com/keras-team/
keras Code used to integrate the software libraries
within software containers, to perform cross
validation, to calculate performance metrics, etc.
are part of the ShinyLearner tool. Source code can
be found at https://github.com/srp33/
ShinyLearner. Data and code used to execute this
analysis are available at https://osf.io/fv8td/. This
repository contains raw and summarized versions
of the analysis results, as well as code that we used
to generate the figures and tables for this
manuscript. The repository is freely available under
the Creative Commons Universal 1.0 license. All
other data are within the manuscript and its
Supporting Information files.

**Funding:** NPG was funded by a student fellowship
from the Simmons Center for Cancer Research at
Brigham Young University. The funders had no role
in study design, data collection and analysis,
decision to publish, or preparation of the
manuscript.

**Competing interests:** The authors have declared
that no competing interests exist.

influence disease development, progression, and/or treatment responses. Patterns that differ between patients in distinct groups (for example, patients who do or do not have a disease or do or do not respond to a treatment) may be used to classify future patients into these groups. This study is a large-scale benchmark comparison of algorithms that can be used to perform such classifications. Additionally, we evaluated feature-selection algorithms, which can be used to identify which variables (genes and/or patient characteristics) are most relevant for classification. Through a series of analyses that build on each other, we show that classification performance varies considerably, depending on which algorithms are used, whether feature selection is used, which settings are used when executing the algorithms, and which metrics are used to evaluate the algorithms' performance. Researchers can use these findings as a resource for deciding which algorithms and settings to prioritize when deriving transcriptome-based biomarkers in future efforts.

## Introduction

Researchers use observational data to derive categories, or classes, into which patients can be assigned. Such classes might include patients who have a given disease subtype, patients at a particular disease stage, patients who respond to a particular treatment, patients who have poor outcomes, patients who have a particular genomic lesion, etc. Subsequently, a physician may use these classes to tailor patient care, rather than using a one-size-fits-all approach[1–3]. However, physicians typically do not know in advance which class labels are most relevant for each patient. Thus, a key challenge is defining objective and reliable criteria for assigning individual patients to known class labels. When such criteria have been identified and sufficiently validated, they can be used in medical "expert systems" for classifying individual patients[4].

In this study, we focused on using gene-expression profiles to perform classification. Gene-expression profiling technologies are relatively mature and are used widely in research[5,6]. In addition, gene-expression profiling is now used in clinical applications. For example, physicians use the *PAM50* classifier, based on the expression of 58 genes, to assign breast-cancer patients to "intrinsic subtypes"[7–11]. The success of this classifier has motivated additional research. In breast cancer alone, more than 100 gene-expression profiles have been proposed for predicting breast-cancer prognosis[12].

Classification algorithms learn from data much as a physician does—past observations inform decisions about new patients. Thus, the first step in developing a gene-expression biomarker is to profile a patient cohort that represents the population of interest. Alternatively, a researcher might use publicly available data for this step. Second, the researcher performs a preliminary evaluation of the potential to assign patients to a particular clinically relevant class based on gene-expression profiles and accompanying clinical information. Furthermore, the researcher may undergo an effort to select a classification algorithm that will perform relatively well for this particular task. Such efforts may be informed by prior experience, a literature review, or trial and error. Using some form of subsampling[13] and a given classification algorithm, the researcher derives a classification model from a subset of the patients' data (training data); to derive this model, the researcher exposes the classification algorithm to the true class labels for each patient. Then, using a disjoint subset of patient observations for which the true class labels have been withheld (test data), the model predicts the label of each patient. Finally, the researcher compares the predictions against the true labels. If the predictive performance approaches or exceeds what can be attained using currently available models, the researcher may continue to refine and test the model. Such steps might include tuning the algorithm,

reducing the number of predictor variables, and testing it on multiple, independent cohorts. In this study, we focus on the need to select algorithm(s).

Modern, high-throughput technologies can produce more than 10,000 gene-expression measurements per biological sample. Thus instead of a traditional approach that uses prior knowledge to determine which genes are included in a predictive model, researchers can use a data-driven approach to infer which genes are most relevant and to identify expression patterns that differ among patient groups[14]. These patterns may be complex, representing subtle differences in expression that span many genes[15]. Due to dependencies among biomolecules and limitations in measurement technologies, high-throughput gene-expression measurements are often redundant and noisy[16]. Thus, to be effective at inferring relevant patterns, classification algorithms must be able to overcome these challenges. One approach is to perform *feature selection* using algorithms that identify predictor variables (features) that are most relevant to the class of interest.

Many machine-learning algorithms and algorithmic variants have been developed and are available in open-source software packages. These include classification algorithms as well as feature-selection algorithms. Gene-expression datasets are abundant in public repositories, affording opportunities for large-scale benchmark comparisons. Furthermore, many of these datasets are accompanied by clinically oriented predictor variables. To our knowledge, no benchmark study has systematically compared the ability to classify patients using clinical data versus gene-expression data—or combined these two types of data—for a large number of datasets. Moreover, previous benchmarks have not systematically evaluated the benefits of optimizing an algorithm's hyperparameters versus using defaults. We address these gaps with a benchmark study spanning 50 datasets (143 class variables representing diverse phenotypes), 52 classification algorithms (1116 hyperparameter combinations), and 14 feature-selection algorithms. We perform this study in a staged design, comparing the ability to classify patients using gene-expression data alone, clinical data alone, or both data types together. In addition, we evaluate the effects of performing hyperparameter optimization and/or feature selection.

## Results

### General trends

We evaluated the predictive performance of 52 classification algorithms on 50 gene-expression datasets. Across the 50 datasets, we made predictions for a total of 143 class variables. We divided the analysis into 5 stages to assess benefits that might come from including clinical predictors, optimizing an algorithm's hyperparameters, or performing feature selection (Fig 1).

In Analysis 1, we used only gene-expression data as predictors and used default hyperparameters for each classification algorithm. S1 Fig illustrates the performance of these algorithms using area under the receiver operating characteristic curve (AUROC) as a performance metric. As a method of normalization, we ranked the classification algorithms for each combination of dataset and class variable. Two patterns emerged. Firstly, 15 of the 18 top-performing algorithms use kernel functions and/or ensemble approaches. Secondly, although some algorithms performed consistently well overall, they performed quite poorly in some cases. For example, the sklearn/logistic_regression algorithm—which used the LibLinear solver[17], a *C* value of 1.0, and no class weighting—resulted in the best average rank; yet for 7 (4.9%) of the dataset/class combinations, its performance ranked in the bottom quartile. The keras/snn algorithm resulted in the second-best average rank; yet for 4 (2.8%) of dataset/class combinations, its performance ranked in the bottom quartile.

This study focuses primarily on AUROC because it is widely used and accounts for moderate levels of class imbalance. However, the performance rankings differed considerably

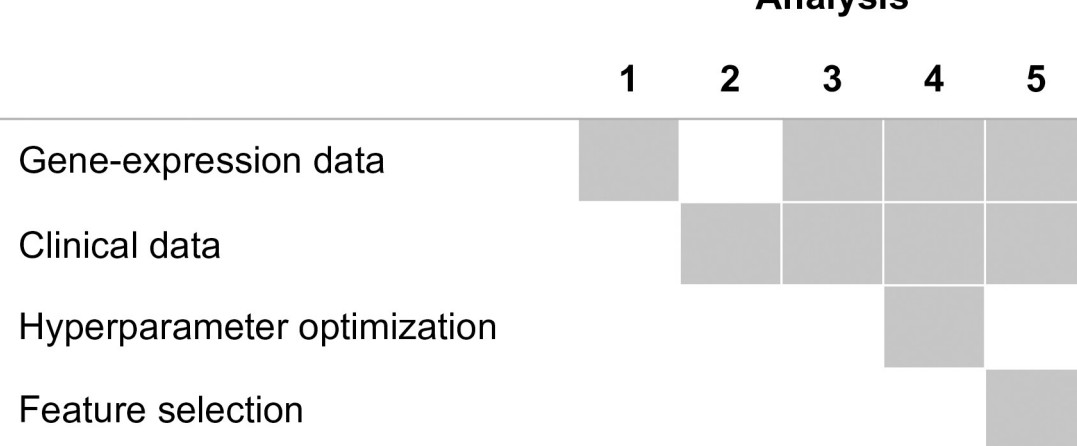

**Fig 1. Overview of analysis scenarios.** This study consisted of five separate but related analyses. This diagram indicates which data type(s) was/were used and whether we attempted to improve predictive performance via hyperparameter optimization or feature selection in each analysis.

depending on which evaluation metric we used. For example, in Analysis 1, many of the same algorithms that performed well according to AUROC also performed well according to classification accuracy (S2 Fig). However, classification accuracy does not account for class imbalance and thus may rank algorithms in a misleading way. For example, the weka/ZeroR algorithm was ranked 18th among the algorithms according to classification accuracy, even though the algorithm simply selects the majority class. (Our analysis included two-class and multi-class problems.) Rankings for the Matthews correlation coefficient were relatively similar to AUROC. For example, sklearn/logistic_regression had the 2nd-best average rank according to this metric. However, in other cases, the rankings were considerably different. For example, the mlr/sda algorithm performed 3rd-best according to MCC but 28th according to AUROC (S3 Fig). The area under the precision-recall curve (AUPRC) is an alternative to the AUROC. In Analysis 1, AUROC and AUPRC scores and ranks were moderately correlated (S4,S5, and S6 Figs). AUPRC is recommended over AUROC when class imbalance is extreme [18,19]. Fig 2 shows the rankings for each algorithm across all metrics that we evaluated, highlighting that conclusions drawn from benchmark comparisons depend heavily on which metric(s) are considered important.

Execution times differed substantially across the algorithms. For Analysis 1, Fig 3 categorizes each algorithm according to its ability to make effective predictions in combination with the computer time required to execute the classification tasks. The sklearn/logistic_regression algorithm not only outperformed other algorithms in terms of predictive ability but also was one of the fastest algorithms. In contrast, the mlr/randomForest algorithm was among the most predictive algorithms but was orders-of-magnitude slower than other top-performing algorithms. Execution time is a less-critical factor than predictive performance; however, when the eventual goal is to provide useful tools for clinical applications, execution times may be an important consideration.

Some classification algorithms are commonly used and thus have been implemented in multiple machine-learning packages. For example, all three open-source libraries that we used in this study have implementations of the SVM and random forests algorithms. However, these implementations differ from each other, often supporting different hyperparameters or using different default values. For example, mlr/svm and weka/LibSVM are both wrappers for

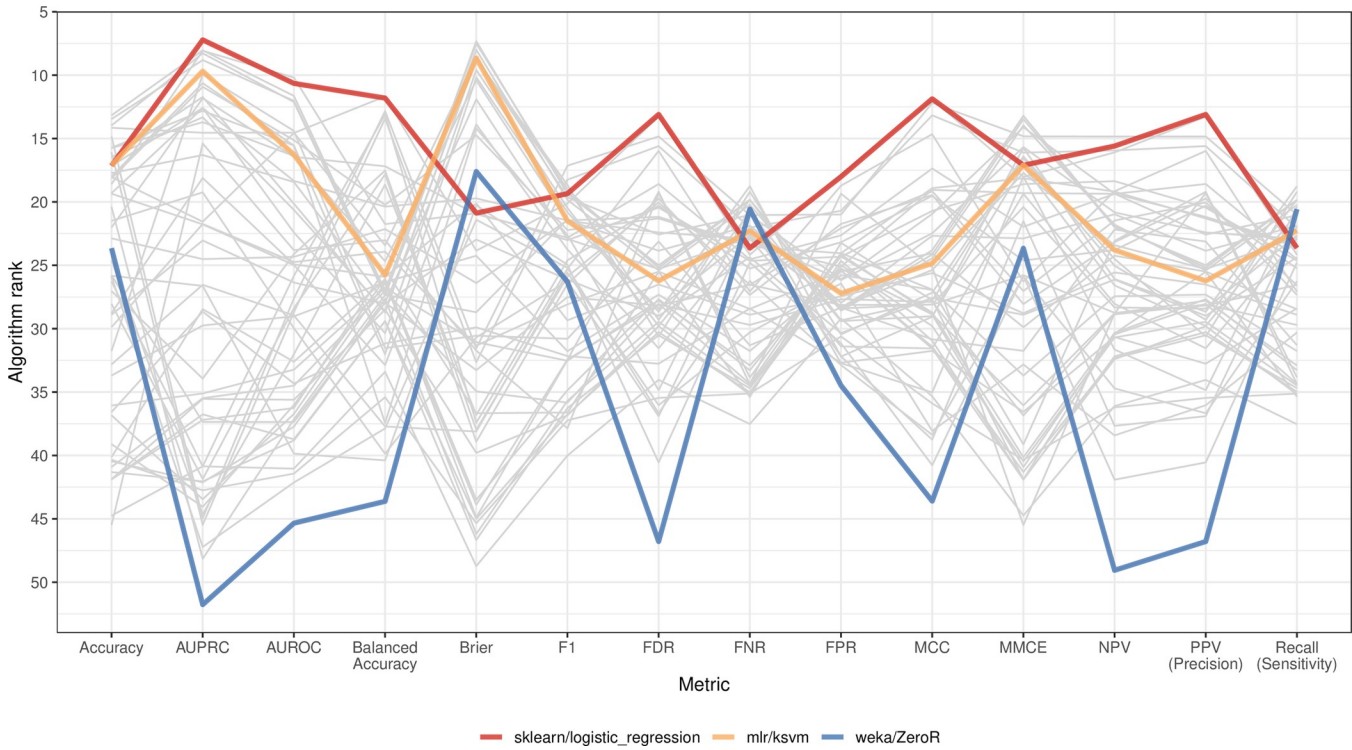

**Fig 2. Comparison of ranks for classification algorithms across performance metrics.** We calculated 14 performance metrics for each classification task. This graph shows results for Analysis 1 (using only gene-expression predictors). For each combination of dataset and class variable, we averaged the metric scores across all Monte Carlo cross-validation iterations. For some metrics (such as Accuracy), a relatively high value is desirable, whereas the opposite is true for other metrics (such as FDR). We ranked the classification algorithms such that relatively low ranks indicated more desirable performance for the metrics and averaged these ranks across the dataset/class combinations. This graph illustrates that the best-performing algorithms for some metrics do not necessarily perform optimally according to other metrics. AUROC = area under the receiver operating characteristic curve. AUPRC = area under the precision-recall curve. FDR = false discovery rate. FNR = false negative rate. FPR = false positive rate. MCC = Matthews correlation coefficient. MMCE = mean misclassification error. NPV = negative predictive value. PPV = positive predictive value.

the LibSVM package[20]; both use a value of 1.0 for the *C* parameter and use the *Radial Basis Function* kernel. However, by default, mlr/svm scales numeric values to zero mean and unit variance, whereas weka/LibSVM performs no normalization by default. In Analysis 1, the predictive performance was similar for these different implementations. Their AUROC values were significantly correlated (r = 0.87; CI = 0.82–0.90; p = 2.2e-16). However, in some instances, their performance differed dramatically. For example, when predicting drug responses for dataset GSE20181, weka/LibSVM performed 2$^{nd}$ best, but mlr/svm performed worst among all algorithms. S7 and S8 Figs illustrate, for two representative datasets, that algorithms with similar methodologies often produced similar predictions; but these predictions were never perfectly correlated. Execution times also differed from one implementation to another; for example, the median execution time for weka/LibSVM was 27.9 seconds, but for mlr/svm it was 114.4 seconds. Overall, the median execution times differed significantly across the software packages (Kruskal-Wallis test; p-value = 1.1e-06). Overall, the sklearn algorithms executed faster than algorithms from other packages (Fig 3).

Some classification labels were easier to predict than others. Across the dataset/class combinations in Analysis 1, the median AUROC across all algorithms ranged between 0.44 and 0.97 (S1 Data). For a given dataset/class combination, algorithm performance varied considerably, though this variation was influenced partially by the weka/ZeroR results, which we used as controls. To gain insight into predictive performance for different types of class labels, we

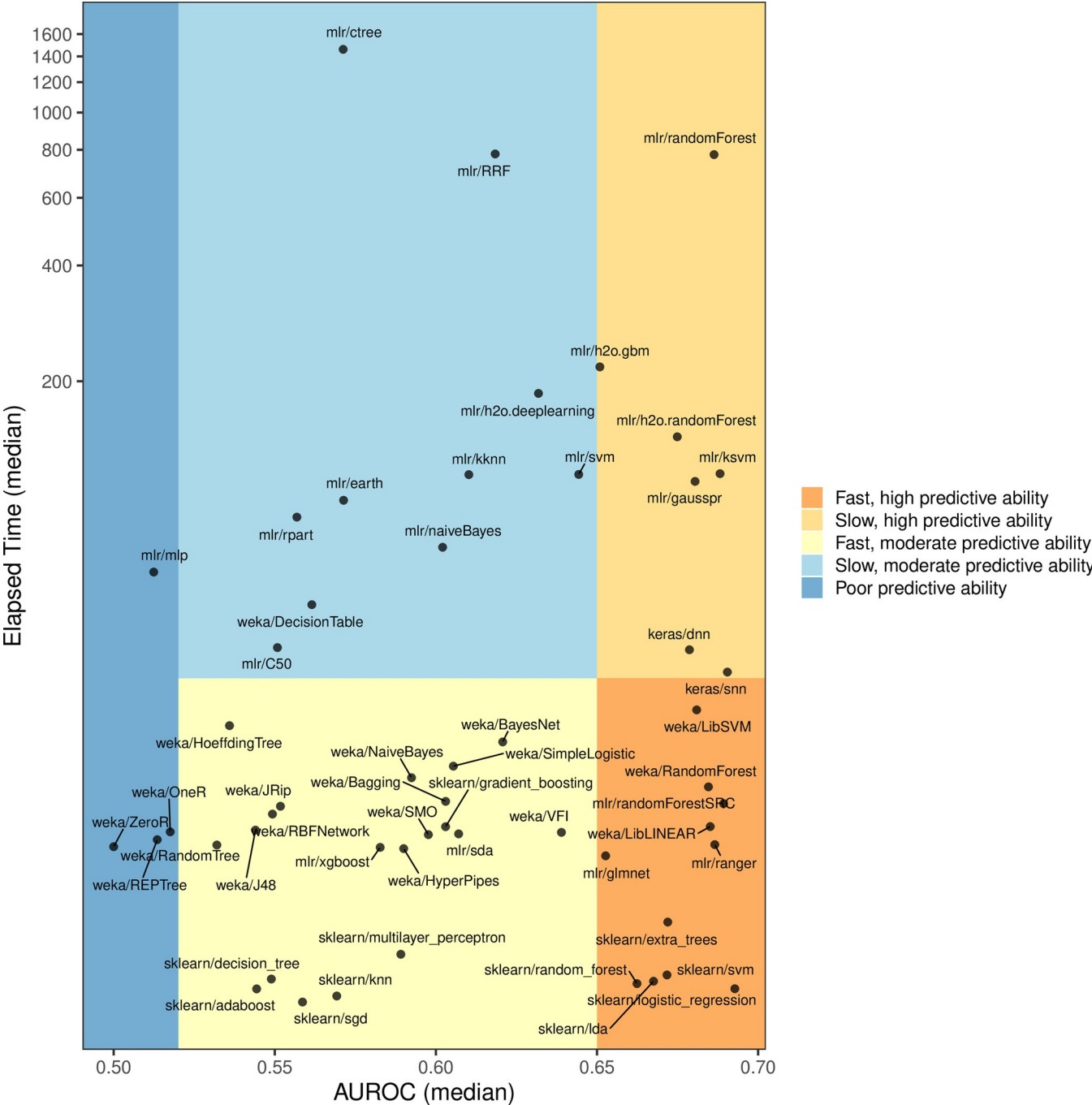

**Fig 3. Tradeoff between execution time and predictive performance for classification algorithms.** When using gene-expression predictors only (Analysis 1), we calculated the median area under the receiver operating characteristic curve (AUROC) across 50 iterations of Monte Carlo cross validation for each combination of dataset, class variable, and classification algorithm. Simultaneously, we measured the median execution time (in seconds) for each algorithm across these scenarios. sklearn/logistic_regression attained the top predictive performance and was the 4th fastest algorithm (median = 5.3 seconds). The coordinates for the y-axis have been transformed to a log-10 scale. We used arbitrary AUROC thresholds to categorize the algorithms based on low, moderate, and high predictive ability.

assigned a category to each class variable (S9 Fig); the best predictive performance was attained for class variables representing molecular markers, histological statuses, and diagnostic labels. Class variables in the "patient characteristics" category performed worst; these variables

represented miscellaneous factors such as the patient's family history of cancer, whether the patient had been diagnosed with multiple tumors, and the patient's physical and cognitive "performance status" at the time of diagnosis.

### Effects of using gene-expression predictors, clinical predictors, or both

In Analysis 2, we used only clinical predictors (for the dataset / class-variable combinations with available clinical data). Three linear-discriminant classifiers performed particularly well: mlr/sda, sklearn/lda, and mlr/glmnet (S10 Fig). Two Naïve Bayes algorithms also ranked among the top performers, whereas these algorithms had performed poorly in Analysis 1. Only two kernel-based algorithms were ranked among the top 10: weka/LibLINEAR and sklearn/logistic_regression. Both of these algorithms use the LibLINEAR solver. Most of the remaining kernel-based algorithms were among the worst performers. As with Analysis 1, most ensemble-based algorithms ranked in the top 25; however, none ranked in the top 5.

S2 Data shows the performance of each combination of dataset and class variable in Analysis 2. As with Analysis 1, the ability to predict particular classes and categories varied considerably (S11 Fig). For approximately two-thirds of the dataset/class combinations, AUROC values decreased—sometimes by more than 0.3 (Fig 4A); however, in a few cases, predictive performance increased. The most dramatic improvement was for GSE58697, in which we predicted progression-free survival for desmoid tumors. The clinical predictors were age at

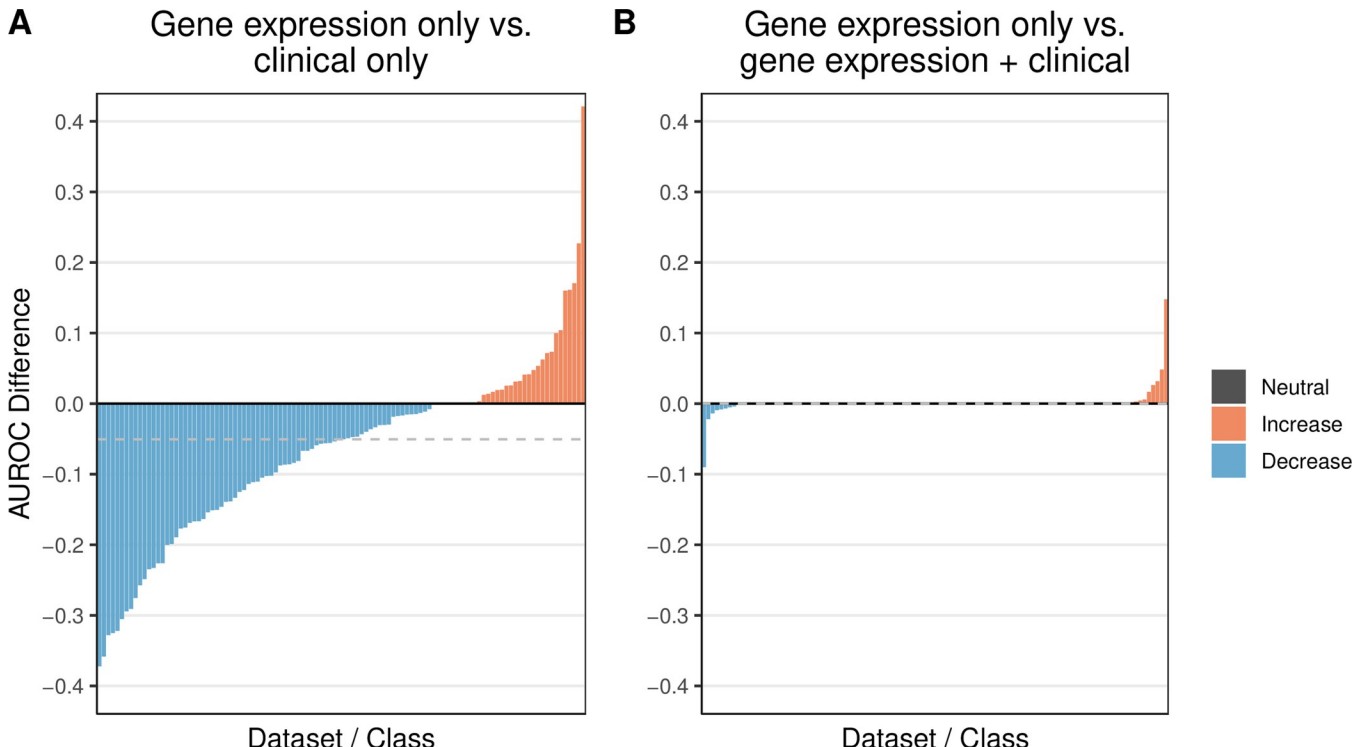

**Fig 4. Relative predictive performance when training on gene-expression predictors alone vs. using clinical predictors alone or gene-expression predictors in combination with clinical predictors.** In both **A** and **B**, we used as a baseline the predictive performance that we attained using gene-expression predictors alone (Analysis 1). We quantified predictive performance using the area under the receiver operating characteristic curve (AUROC). In **A**, we show the relative increase or decrease in performance when using clinical predictors alone (Analysis 2). In most cases, AUROC values decreased; however, in a few cases, AUROC values increased (by as much as 0.42). In **B**, we show the relative change in performance when using gene-expression predictors in combination with clinical predictors (Analysis 3). For 82/109 (75%) of dataset/class combinations, including clinical predictors had no effect on performance. However, for the remaining combinations, the AUROC improved by as much as 0.15 and decreased by as much as 0.09.

diagnosis, biological sex, and tumor location. Salas, et al. previously found in a univariate analysis that age at diagnosis was significantly correlated with progression-free survival [21]. We focused on patients who experienced relatively long or short survival times and used multivariate methods.

In Analysis 3, we combined clinical and gene-expression predictors. We limited this analysis to the 108 dataset / class-variable combinations for which clinical predictors were available (S3 Data and S12 Fig). As with Analysis 1, kernel- and ensemble-based algorithms performed best overall (S13 Fig). For 90 (83.3%) of the dataset/ class-variable combinations, the AUROC values were identical to Analysis 1 (Fig 4B). Except in three cases, the absolute change in AUROC was smaller than 0.05, including for GSE58697 (0.026 increase). These results suggest that standard classification algorithms (using default parameters) may not be well suited to datasets in which gene-expression and clinical predictors have simply been merged. The abundance of gene-expression variables may distract the algorithms and/or obfuscate signal from the relatively few clinical variables. Additionally, gene-expression and clinical predictors may carry redundant signals.

## Effects of performing hyperparameter optimization

In Analysis 4, we performed hyperparameter optimization via nested cross validation. Across all 52 classification algorithms, we employed 1116 distinct hyperparameter combinations under the assumption that the default settings may be suboptimal for the datasets we evaluated. When clinical predictors were available, we included them (as in Analysis 3). When no clinical predictors were available, we used gene-expression data only (as in Analysis 1). Again, kernel- and ensemble-based algorithms performed well overall (S14 Fig), although the individual rankings differed modestly from the previous analyses. The weka/LibLINEAR algorithm had the best median rank, and algorithms based on random forests were generally ranked lower than in previous analyses. For most dataset / class-variable combinations, the AUROC (median across all classification algorithms) improved with hyperparameter optimization (Fig 5A); however, in some cases, performance decreased.

The best- and worst-performing class variables and categories were similar to the previous analyses (S15 Fig and S4 Data). We observed a positive trend in which datasets with larger sample sizes resulted in higher median AUROC values (S16 Fig); however, this relationship was not statistically significant (Spearman's rho = 0.13; p = 0.13). We observed a slightly negative trend between the number of genes in a dataset and median AUROC (S17 Fig), but again this relationship was not statistically significant (rho = -0.07; p = 0.43).

Evaluating many hyperparameter combinations enabled us to quantify how much the predictive performance varied for different combinations. Some variation is desirable because it enables algorithms to adapt to diverse analysis scenarios; however, large amounts of variation make it difficult to select hyperparameter combinations that are broadly useful. For some classification algorithms, AUROC values varied widely across hyperparameter combinations when applied to a given dataset / class variable (S18 Fig). These variations were often different for algorithms with similar methodological approaches. For example, the median coefficient of variation was 0.22 for the sklearn/svm algorithm but 0.08 for mlr/svm and 0.06 for weka/LibSVM. In other cases, AUROC varied little across hyperparameter combinations. For example, the four algorithms with the highest median AUROC—weka/LibLINEAR, mlr/glmnet, sklearn/logistic_regression, and sklearn/extra_trees—had median coefficients of variation of 0.02, 0.03, 0.01, and 0.03, respectively. For each of these algorithms, we plotted the performance of all hyperparameter combinations across all dataset / class-variable combinations (S19, S20, S21, and S22 Figs). The default hyperparameter combination failed to perform best

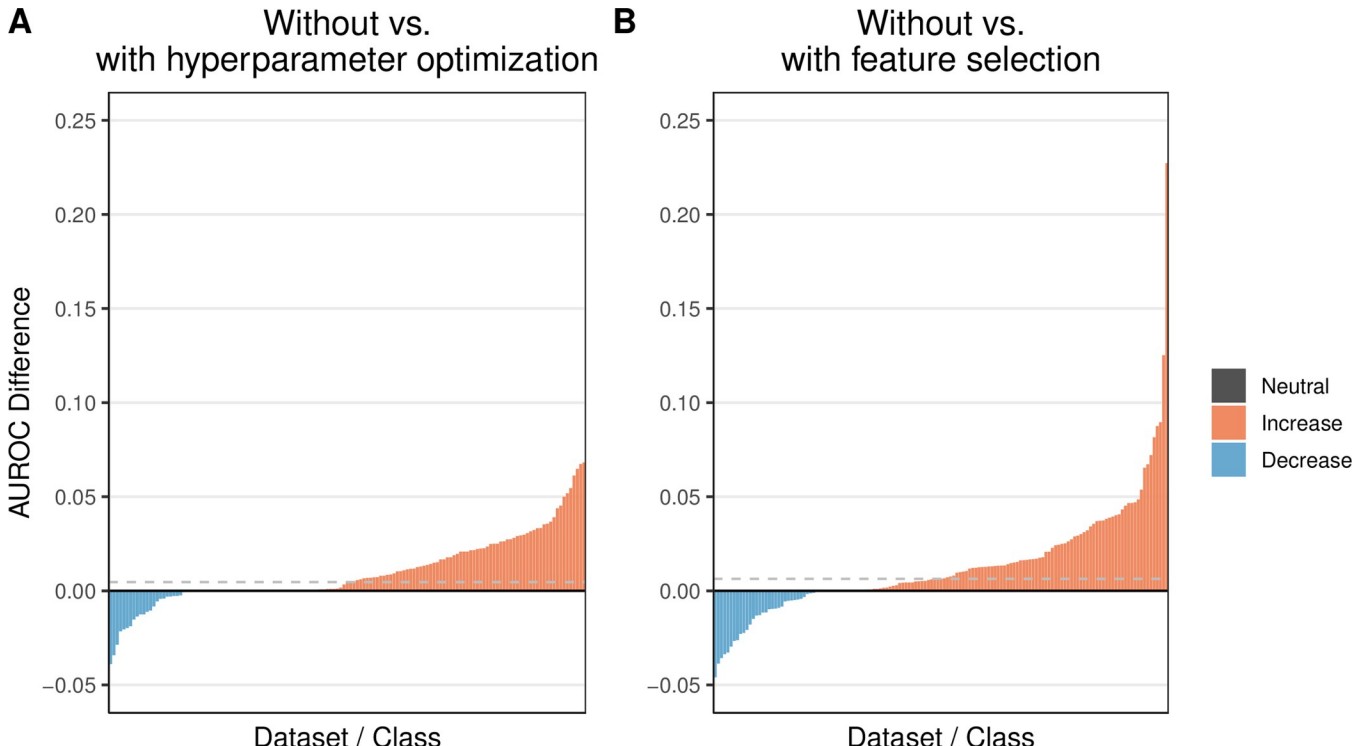

**Fig 5. Relative predictive performance when using default algorithm hyperparameters and all features vs. tuning hyperparameters or selecting features.** In both **A** and **B**, we use as a baseline the predictive performance that we attained using default hyperparameters for the classification algorithms (Analysis 3). We quantified predictive performance using the area under the receiver operating characteristic curve (AUROC). In **A**, we show the relative increase or decrease in performance when tuning hyperparameters within each training set (Analysis 4). In most cases, AUROC values increased. In **B**, we show the relative change in performance when performing feature selection within each training set (Analysis 5). AUROC increased for most dataset / class-variable combinations. The horizontal dashed lines indicate the median improvement across all dataset / class-variable combinations.

for any of these algorithms. Indeed, for two of the four algorithms, the default combination performed *worst*.

Of the 1116 total combinations, 1084 were considered best for at least one dataset / class-variable combination (based on average performance in inner cross-validation folds).

### Effects of performing feature selection

In Analysis 5, we performed feature selection via nested cross validation. We used 14 feature-selection algorithms in combination with each classification algorithm. Due to the computational demands of evaluating these 728 combinations, we initially used default hyperparameters for both types of algorithms. The feature-selection algorithms differed in their methodological approaches (Table 1). In addition, some were univariate methods, while others were multivariate. Some feature-selection algorithms mirrored the behavior of classification algorithms (e.g., SVMs or random forests); others were based on statistical inference or entropy-based metrics.

Once again, kernel- and ensemble-based classification algorithms performed best overall when feature selection was used (Fig 6). The median improvement per dataset / class-variable combination was slightly larger for feature selection than for hyperparameter optimization, and the maximal gains in predictive performance were larger for feature selection (Fig 5B and S5 Data). Overall, there was a strong positive correlation between AUROC values for Analyses 4 and 5 (Spearman's rho = 0.73; S23 Fig). Among the 10 dataset / class-variable combinations

**Table 1. Summary of feature-selection algorithms.** We evaluated 14 feature-selection algorithms. The abbreviation for each algorithm contains a prefix that indicates which machine-learning library implemented the algorithm (mlr = Machine learning in R, sklearn = scikit-learn, weka = WEKA: The workbench for machine learning). For each algorithm, we provide a brief description of the algorithmic approach; we extracted these descriptions from the libraries that implemented the algorithms. In addition, we assigned high-level categories that indicate whether the algorithms evaluate a single feature (univariate) or multiple features (multivariate) at a time. In some cases, the individual machine-learning libraries aggregated algorithm implementations from third-party packages. In these cases, we cite the machine-learning library and the third-party package. When available, we also cite papers that describe the algorithmic methodologies used.

| Abbreviation | Description | Category |
|---|---|---|
| mlr/cforest.importance | Uses the permutation principle (based on Random Forests) to calculate standard and conditional importance of features[22–24] | Multivariate |
| mlr/kruskal.test | Uses the Kruskal-Wallis rank sum test[22,25] | Univariate |
| mlr/randomForestSRC.rfsrc | Uses the error rate for trees grown with and without a given feature[22,26,27] | Multivariate |
| mlr/randomForestSRC.var.select | Selects variables using minimal depth (Random Forests)[22,26,27] | Multivariate |
| sklearn/mutual_info | Calculates the mutual information between two feature clusterings[28,29] | Univariate |
| sklearn/random_forest_rfe | Recursively eliminates features based on Random Forests classification[28,30] | Multivariate |
| sklearn/svm_rfe | Recursively eliminates features based on support vector classification[28,31] | Multivariate |
| weka/Correlation | Calculates Pearson's correlation coefficient between each feature and the class[32,33] | Univariate |
| weka/GainRatio | Measures the gain ratio of a feature with respect to the class[32,34] | Univariate |
| weka/InfoGain | Measures the information gain of a feature with respect to the class[32,34] | Univariate |
| weka/OneR | Evaluates the worth of a feature using the OneR classifier[32,35] | Univariate |
| weka/ReliefF | Repeatedly samples an instance and considers the value of a given attribute for the nearest instance of the same and different class[32,36] | Multivariate |
| weka/SVMRFE | Recursively eliminates features based on support vector classification[31,32] | Multivariate |
| weka/SymmetricalUncertainty | Measures the symmetrical uncertainty of a feature with respect to the class[32,37] | Univariate |

that improved most after feature selection, 8 were associated with prognostic, stage, or patient-characteristic variables—categories that were most difficult to predict overall (S24 Fig). The remaining two combinations were molecular markers (HER2-neu and progesterone receptor status). Generally, the best performance was attained using 100 or 1000 features (S25 Fig).

Across all classification algorithms, the weka/Correlation feature-selection algorithm resulted in the best predictive performance (S26 Fig), despite being a univariate method. This algorithm calculates the Pearson's correlation coefficient between each feature and the class values, a relatively simple approach that also ranked among the fastest (S27 Fig). Other univariate algorithms were among the top performers. To characterize algorithm performance further, we compared the feature ranks between all algorithm pairs for two of the datasets. Some pairs produced highly similar gene rankings, whereas in other cases the similarity was low (S28 and S29 Figs). The weka/Correlation and mlr/kruskal.test algorithms produced similar feature ranks; both use statistical inference; the former is a parametric method, while the latter is nonparametric.

Some classification algorithms (e.g., weka/ZeroR and sklearn/decision_tree) performed poorly irrespective of feature-selection algorithm, whereas other classification algorithms (e.g., mlr/ranger and weka/LibLINEAR) performed consistently well across feature-selection algorithms (S30 Fig). The performance of other algorithms was more variable.

To provide guidance to practitioners, we examined interactions between individual feature-selection algorithms and classification algorithms (Fig 7). If a researcher had identified a particular classification algorithm to use, they might wish to select a feature-selection algorithm that performs well in combination with that classification algorithm. For example, the weka/Correlation feature-selection algorithm performed best overall, but it was only the 6th-best algorithm on average when sklearn/logistic_regression was used for classification. In contrast, a feature-selection algorithm that underperforms in general may perform well in combination

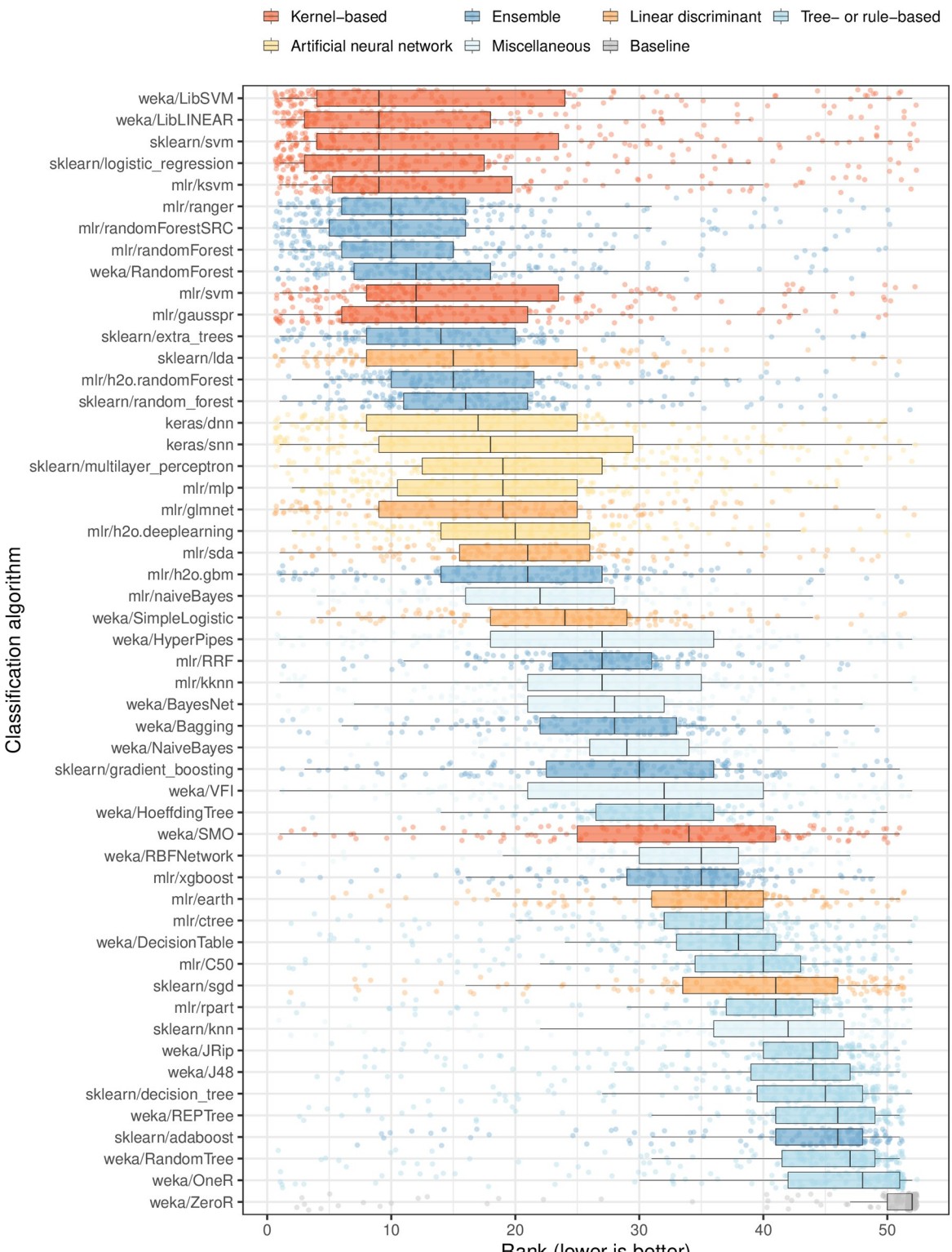

**Fig 6. Relative performance of classification algorithms using gene-expression and clinical predictors and performing feature selection.** We predicted patient states using gene-expression and clinical predictors with feature selection (Analysis 5). We used nested cross validation to estimate which features would be optimal for each algorithm in each training set. For each combination of dataset, class variable, and classification algorithm, we calculated the arithmetic mean of area under the receiver operating characteristic curve (AUROC) values across 5 iterations of Monte Carlo cross-validation. Next, we sorted the algorithms based on the average rank across all dataset/class combinations. Each data point that overlays the box plots represents a particular dataset/class combination.

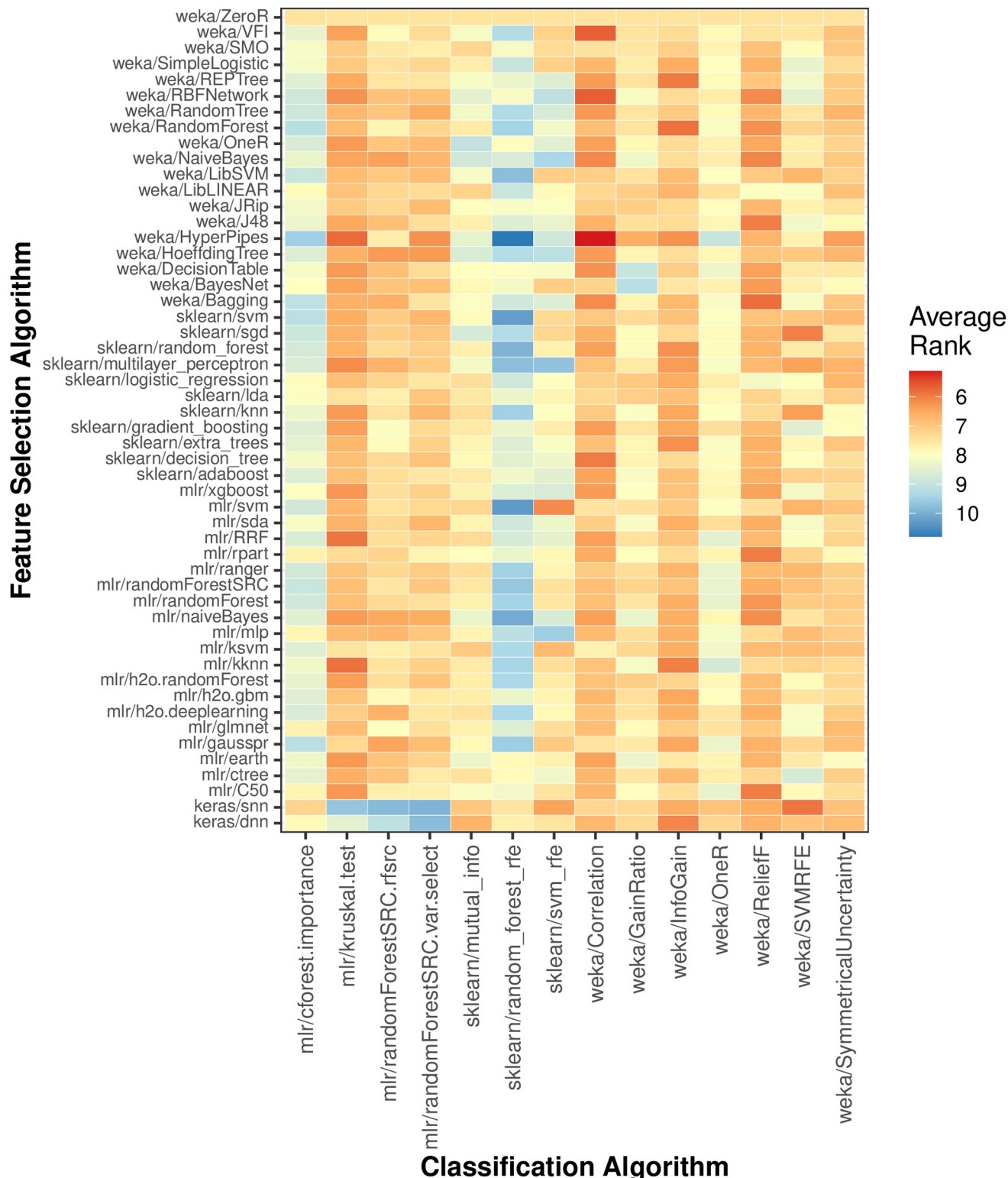

**Fig 7. Relative classification performance per combination of feature-selection and classification algorithm.** For each combination of dataset and class variable, we averaged area under receiver operating characteristic curve (AUROC) values across all Monte Carlo cross-validation iterations. Then for each classification algorithm, we ranked the feature-selection algorithms based on AUROC scores across all datasets and class variables. Lower ranks indicate better performance. Dark-red boxes indicate cases where a particular feature-selection algorithm was especially effective for a particular classification algorithm. The opposite was true for dark-blue boxes.

with a given classification algorithm. For example, sklearn/svm_rfe performed poorly overall but was effective in combination with mlr/svm.

We evaluated two alternatives for performing feature selection. Firstly, for 5 dataset/class combinations and 7 feature-selection algorithms, we used hyperparameter combinations for the feature-selection algorithms that differed from the defaults (a total of 59 hyperparameter combinations). The results were similar to Analysis 5 (S31 Fig and S6 Data), and the median change in AUROC per dataset/class combination was a decrease of 0.007. Secondly, all of the feature-selection algorithms are filter based; thus, ranking is performed independently of classification. As an alternative, wrapper-based approaches evaluate the extent to which features improve classification performance. We evaluated two classification algorithms (sklearn/svm and sklearn/knn) and selected the top 0.01%, 0.1% or 1% of features. The median change in AUROC per dataset/class combination was a decrease of 0.011. Additional benchmarks involving more algorithms and datasets are warranted in future studies.

Finally, we note that feature selection can be used to provide biological insight. Features that are consistently ranked highly for a given disease may be more likely to play a role in disease development or progression. For GSE10320 and GSE46691, we identified the 50 top-ranked genes, averaged across all algorithms (S7 Data), and used the Molecular Signatures Database to quantify the overlap between these gene lists and a curated "hallmark" set of gene sets known to play a role in tumorigenesis[38]. Three and four gene sets, respectively, significantly overlapped with the top-ranked genes (S8 and S9 Data). More extensive analysis and lab work would be required to validate these insights.

## Discussion

The machine-learning community has developed hundreds of classification algorithms, spanning diverse methodological approaches[39]. Historically, most datasets available for testing had fewer than 100 predictor variables, so most algorithms were created and optimized for that use case[40]. Consequently, the execution time and predictive performance of many classification algorithms may be unsatisfactory when datasets consist of thousands of predictor variables–the algorithms may have difficulty identifying the most informative features in the data[41,42].

This benchmark study is considerably larger than any prior study of classification algorithms applied to gene-expression data. When gene-expression microarrays became common in biomedical research in the early 2000s, researchers began exploring the potential to make clinically relevant predictions and overcome these challenges[43–47]. As a result of data-sharing policies, gene-expression datasets were increasingly available in the public domain, and researchers performed benchmark studies, comparing the effectiveness of classification algorithms on gene-expression data[14,48–50]. Each of these studies evaluated between 5 and 21 algorithmic variants. In addition, the authors typically used at least one method of feature selection to reduce the number of predictor variables. The studies used as many as 7 datasets, primarily from tumor cells (and often adjacent normal cells). The authors focused mostly on classical algorithms, including k-Nearest Neighbors[51], linear discriminant analysis[52], and the multi-layer perceptron[53]. Pochet, et al. also explored the potential for nonlinear Support Vector Machine (SVM) classifiers to increase predictive performance relative to linear methods[49,54]. Later benchmark studies highlighted two types of algorithm—SVM and random forests[30]—that performed relatively well on gene-expression data[42,55–57]. Statnikov, et al. examined 22 datasets and specifically compared the predictive capability of these two algorithm types. Overall, they found that SVMs significantly outperformed random forests, although random forests outperformed SVMs in some cases[42]. Perhaps in part due to these

highly cited studies, SVMs and random forests have been used heavily in diverse types of biomedical research over the past two decades[58].

Community efforts—especially the Sage Bionetworks DREAM Challenges and Critical Assessment of Massive Data Analysis challenges[59–61]—have encouraged the development and refinement of predictive models to address biomedical questions. In these benchmark studies, the priority is to maximize predictive performance and thus increase the potential that the models will have in practical use. Accordingly, participants have flexibility to use alternative normalization or summarization methods, to use alternative subsets of the training data, to combine algorithms, etc. These strategies often prove useful; however, this heterogeneity makes it difficult to deconvolve the relationship between a given solution's performance and the underlying algorithm(s), hyperparameters, and features used.

Our primary motivation is to provide helpful advice for practitioners who perform biomarker studies. Identifying algorithm(s) and hyperparameter(s) that perform consistently well in this setting may ultimately lead to patient benefits. In situations where a biomarker is applied to thousands of cancer patients, even modest increases in accuracy can benefit hundreds of patients. Accordingly, we questioned whether SVM and random forests algorithms would continue to be the top performers when compared against diverse types of classification algorithms. We also questioned whether there would be scenarios in which these algorithms would perform poorly. Furthermore, relatively little has been known about the extent to which algorithm choice affects predictive success for a given dataset. Thus, we questioned how much variance in predictive performance we would see across the algorithms. In addition, we evaluated practical matters such as tradeoffs between predictive performance and execution time, the extent to which algorithm rankings are affected by the performance metric used, and which algorithms behave most similarly—or differently—to each other.

Our secondary motivation was to help bridge the gap between machine-learning researchers who develop general-purpose algorithms and biomedical researchers who seek to apply them in a specific context. When selecting algorithm(s), hyperparameters, and features to use in a biomarker study, researchers might base their decisions on what others have reported in the literature for a similar study; or they might consider anecdotal experiences that they or their colleagues have had. However, these decisions may lack an empirical basis and not generalize from one analysis to another. Alternatively, researchers might apply many algorithms to their data to estimate which algorithm(s) will perform best. However, this approach is time- and resource-intensive and may lead to bias if the comparisons are not performed in a rigorous manner. In yet another approach, researchers might develop a custom classification algorithm, perhaps one that is specifically designed for the target data. However, it is difficult to know whether such an algorithm would outperform existing, classical algorithms.

Many factors can affect predictive performance in a biomarker study. These factors include data-generation technologies, data normalization / summarization processes, validation strategies, and evaluation metrics used. Although such factors must be considered, we have shown that when holding them constant, the choice of algorithm, hyperparameter combination, and features usually affects predictive performance for a given dataset—sometimes dramatically. Despite these variations, we have demonstrated that particular algorithms and algorithm categories consistently outperform others across diverse gene-expression datasets and class variables. However, even the best algorithms performed poorly in some cases. These findings support the theory that no single algorithm is universally optimal[62]. But they also suggest that researchers can increase the odds of success in developing accurate biomarkers by focusing on a few top-performing algorithms and using hyperparameter optimization and/or feature selection, despite the additional computational demands in performing these steps. However, it is subjective to decide which characteristics to optimize and whether such optimization will reap rewards.

We deliberately focused on general-purpose algorithms because they are readily available in well-maintained, open-source packages. Of necessity, we evaluated an inexhaustive list of algorithms and hyperparameter combinations. Other algorithms or hyperparameter combinations may have performed better than those that we used. Many studies have proposed algorithm variations designed for feature selection and/or classification of gene-expression data[63–72]. Some algorithms in our study had more hyperparameter combinations than others, which may have enabled those algorithms to adapt better in Analysis 4. Additionally, in some cases, our hyperparameter combinations were inconsistent between two algorithms of the same type because different software libraries support different options. Despite these limitations, a key advantage of our benchmarking approach is that we performed these comparisons in an impartial manner, not having developed any of the algorithms that we evaluated nor having other conflicts of interest that might bias our results.

Generally, kernel- and ensemble-based algorithms outperformed other types of algorithms in our analyses. Other algorithm types—such as linear-discriminant and neural-network algorithms—performed well in some scenarios. Deep neural networks have received much attention in the biomedical literature over the past decade[73]. This study included three types of deep neural networks. keras/snn and keras/dnn used fully connected networks; the hyperparameters combinations differed in the number of nodes, number of layers, dropout rate, regularization rate, number of epochs, and whether batch normalization was used. The mlr/h2o. deeplearning algorithm provided many of the same options. In Analysis 1, the keras/snn andn keras/dnn algorithms ranked among the top 11; however, their performance dropped in subsequent analyses. The mlr/h2o.deeplearning algorithm performed at mediocre levels in all of our analyses. Custom adaptations to this (or any other) deep-learning algorithm may improve predictive performance in future studies. Efforts to improve predictive ability might also include optimizing hyperparameters of feature-selection algorithms, combining hyperparameter-optimized classification algorithms with feature selection, and using multiple classifier systems [74]. Transfer learning across datasets may also prove fruitful[75].

Our findings are specific to high-throughput gene-expression datasets that have either no clinical predictors or a small set of clinical predictors. However, our conclusions may have relevance to other datasets that include many features and that include a combination of numeric and categorical features.

We applied Monte Carlo cross validation to each dataset separately and thus did not evaluate predictive performance on independent datasets. This approach was suitable for our benchmark comparison because our priority was to compare algorithms against each other rather than to optimize their performance for clinical use. On another note, comparisons across machine-learning packages are difficult to make. For example, some *sklearn* algorithms provided the ability to automatically address class imbalance, whereas other software packages did not always provide this functionality. Adapting these weights manually was infeasible for this study. Accordingly, future research that specifically focuses on under-sampling, over-sampling, and other methods to correct for class imbalance is warranted. In addition, some classification algorithms are designed to produce probabilistic predictions, whereas other algorithms produce only discrete predictions. The latter algorithms may have been at a disadvantage in our benchmark for the AUROC and other metrics.

## Methods

### Ethics statement

Brigham Young University's Institutional Review Board approved this study under exemption status. This study uses data collected from public repositories only. We played no part in patient recruiting or in obtaining consent.

## Data preparation

We used 50 datasets spanning diverse diseases and tissue types but focused primarily on cancer-related conditions. We used data from two sources. The first was a resource created by Golightly, et al.[76] that includes 45 datasets from Gene Expression Omnibus[77]. For these datasets, the gene-expression data were generated using Affymetrix microarrays, normalized using Single Channel Array Normalization[78], summarized using BrainArray annotations [79], quality checked using IQRay[80] and DoppelgangR[81], and batch-adjusted (where applicable) using ComBat[82]. Depending on the Affymetrix platform used, expression levels were available for 11,832 to 21,614 genes. For the remaining 5 datasets, we used RNA-Sequencing data from The Cancer Genome Atlas (TCGA)[83], representing 5 tumor types: colorectal adenocarcinoma (COAD), bladder urothelial carcinoma (BLCA), kidney renal clear cell carcinoma (KIRC), prostate adenocarcinoma (PRAD), and lung adenocarcinoma (LUAD). These data had been aligned and quantified using the Rsubread and featureCounts packages[84,85], resulting in transcripts-per-million values for 22,833 genes[86]. All gene-expression data were labeled using Ensembl gene identifiers[87].

For the microarray datasets, we used the class variables and clinical variables identified by Golightly, et al. (2.8 class variables per dataset)[76]. For the RNA-Sequencing datasets, we identified a total of 16 class variables. When a given sample was missing data for a given class variable, we excluded that sample from the analyses. Some class variables were continuous in nature (e.g., overall survival). We discretized these variables to enable classification, taking into account censor status where applicable. To support consistency and human interpretability across datasets, we assigned a standardized name and category to each class variable; the original and standardized names are available in S10 Data.

For most of the Golightly, et al. datasets, at least one clinical variable had been identified as a potential predictor variable. For TCGA datasets, we selected multiple clinical-predictor variables per dataset. Across all datasets, the mean and median number of clinical predictors per dataset were 3.1 and 2.0, respectively (S10 Data). We avoided combinations of clinical-predictor variables and class variables that were potentially confounded. For example, when a dataset included cancer stage as a class variable, we excluded predictor variables such as tumor grade or histological status. In some cases, no suitable predictor variable was available for a given class variable, leaving only gene-expression variables as predictors; this was true for 35 class variables.

## Algorithms used

We used 52 classification algorithms that were implemented in the ShinyLearner tool, which enables researchers to benchmark algorithms from open-source machine-learning libraries and is redistributed as software containers[88,89]. We used implementations from the *mlr* R package (version 2; R version 3.5)[22], *sklearn* Python module (versions 0.18–0.22)[28], *Weka* Java application (version 3.6)[32], and *keras* Python module (2.6.0). Table 2 lists each algorithm, along with a description and methodological category for each algorithm. Furthermore, it indicates the open-source software package that implemented the algorithm, as well as the number of unique hyperparameter combinations that we evaluated for each algorithm. A full list can be found in S11 Data. Among the classification algorithms was Weka's *ZeroR*, which predicts all instances to have the majority class. We included this algorithm as a sanity check [90] and a baseline against which all other algorithms could be compared. Beyond the 52 classification algorithms that we used, additional algorithms were available in ShinyLearner. We excluded algorithms that raised exceptions when we used default hyperparameters, required excessive amounts of random access memory (75 gigabytes or more), or were orders of magnitude slower than the other algorithms.

**Table 2. Summary of classification algorithms.** We compared the predictive ability of 52 classification algorithms that were available in ShinyLearner and had been implemented across 4 open-source machine-learning libraries. The abbreviation for each algorithm contains a prefix indicating which machine-learning library implemented the algorithm (mlr = Machine learning in R, sklearn = scikit-learn, weka = WEKA: The workbench for machine learning; keras = Keras). For each algorithm, we provide a brief description of the algorithmic approach; we extracted these descriptions from the libraries that implemented the algorithms. In addition, we assigned high-level categories that characterize the algorithmic methodology used by each algorithm. In some cases, the individual machine-learning libraries aggregated algorithm implementations from third-party packages. In these cases, we cite the machine-learning library and the third-party package. When available, we also cite papers that describe the algorithmic methodologies used. Finally, for each algorithm, we indicate the number of unique hyperparameter combinations evaluated in Analysis 4.

| Abbreviation | Description | Category | Combos |
|---|---|---|---|
| keras/dnn | Multi-layer neural network with Exponential Linear Unit activation[91,92] | Artificial neural network | 54 |
| keras/snn | Multi-layer neural network with Scaled Exponential Linear Unit activation[91,92] | Artificial neural network | 54 |
| mlr/C50 | C5.0 Decision Trees[22,93] | Tree- or rule-based | 32 |
| mlr/ctree | Conditional Inference Trees[22,94] | Tree- or rule-based | 4 |
| mlr/earth | Multivariate Adaptive Regression Splines[22,95] | Linear discriminant | 36 |
| mlr/gausspr | Gaussian Processes[22,96] | Kernel-based | 3 |
| mlr/glmnet | Generalized Linear Models with Lasso or Elasticnet Regularization[22,97] | Linear discriminant | 3 |
| mlr/h2o.deeplearning | Deep Neural Networks[22,92,98] | Artificial neural network | 32 |
| mlr/h2o.gbm | Gradient Boosting Machines[22,98,99] | Ensemble | 16 |
| mlr/h2o.randomForest | Random Forests[22,30,98] | Ensemble | 12 |
| mlr/kknn | k-Nearest Neighbor[22,100] | Miscellaneous | 6 |
| mlr/ksvm | Support Vector Machines[22,54,96] | Kernel-based | 40 |
| mlr/mlp | Multi-Layer Perceptron[22,53,101] | Artificial neural network | 14 |
| mlr/naiveBayes | Naive Bayes[22,102] | Miscellaneous | 2 |
| mlr/randomForest | Breiman and Cutler's Random Forests[22,103] | Ensemble | 12 |
| mlr/randomForestSRC | Fast Unified Random Forests for Survival, Regression, and Classification[22,26,27] | Ensemble | 108 |
| mlr/ranger | A Fast Implementation of Random Forests[22,104] | Ensemble | 12 |
| mlr/rpart | Recursive Partitioning and Regression Trees[22,105,106] | Tree- or rule-based | 1 |
| mlr/RRF | Regularized Random Forests[22,107] | Ensemble | 24 |
| mlr/sda | Shrinkage Discriminant Analysis[22,108] | Linear discriminant | 2 |
| mlr/svm | Support Vector Machines[20,22,102] | Kernel-based | 28 |
| mlr/xgboost | eXtreme Gradient Boosting[22,109] | Ensemble | 3 |
| sklearn/adaboost | AdaBoost[28,110] | Ensemble | 8 |
| sklearn/decision_tree | A decision tree classifier[28] | Tree- or rule-based | 96 |
| sklearn/extra_trees | An extra-trees classifier[28] | Ensemble | 24 |
| sklearn/gradient_boosting | Gradient Boosting for classification[28,99] | Ensemble | 6 |
| sklearn/knn | k-nearest neighbors vote[28,51] | Miscellaneous | 12 |
| sklearn/lda | Linear Discriminant Analysis[28] | Linear discriminant | 3 |
| sklearn/logistic_regression | Logistic Regression[28,111] | Kernel-based | 32 |
| sklearn/multilayer_perceptron | Multi-layer Perceptron[28,53] | Artificial neural network | 24 |
| sklearn/random_forest | Random Forests[28,30] | Ensemble | 24 |
| sklearn/sgd | Linear classifiers with stochastic gradient descent training[28,112] | Linear discriminant | 36 |
| sklearn/svm | C-Support Vector Classification[28,54] | Kernel-based | 32 |
| weka/Bagging | Bagging a classifier to reduce variance[32,113] | Ensemble | 32 |
| weka/BayesNet | Bayes Network learning using various search algorithms and quality measures[32,114] | Miscellaneous | 2 |
| weka/DecisionTable | Simple decision table majority classifier[32,115] | Tree- or rule-based | 6 |
| weka/HoeffdingTree | Hoeffding tree[32,116] | Tree- or rule-based | 32 |
| weka/HyperPipes | HyperPipe classifier[32] | Miscellaneous | 1 |
| weka/J48 | Pruned or unpruned C4.5 decision tree[32,117] | Tree- or rule-based | 96 |
| weka/JRip | Repeated Incremental Pruning to Produce Error Reduction[32,118] | Tree- or rule-based | 12 |
| weka/LibLINEAR | LIBLINEAR—A Library for Large Linear Classification[17,32] | Kernel-based | 16 |
| weka/LibSVM | Support vector machines[20,32] | Kernel-based | 32 |
| weka/NaiveBayes | A Naive Bayes classifier using estimator classes[32,119] | Miscellaneous | 3 |

*(Continued)*

**Table 2.** (Continued)

| Abbreviation | Description | Category | Combos |
|---|---|---|---|
| weka/OneR | 1R (1 rule) classifier[32,35] | Tree- or rule-based | 3 |
| weka/RandomForest | Forest of random trees[30,32] | Ensemble | 18 |
| weka/RandomTree | Tree that considers K randomly chosen attributes at each node[32] | Tree- or rule-based | 2 |
| weka/RBFNetwork | Normalized Gaussian radial basis function network[32] | Miscellaneous | 18 |
| weka/REPTree | Fast decision tree learner (reduced-error pruning with backfitting)[32] | Tree- or rule-based | 16 |
| weka/SimpleLogistic | Linear logistic regression models[32,120,121] | Linear discriminant | 5 |
| weka/SMO | Sequential minimal optimization for a support vector classifier[32,122–124] | Kernel-based | 20 |
| weka/VFI | Voting feature intervals[32,125] | Miscellaneous | 6 |
| weka/ZeroR | 0-R classifier (predicts the mean for a numeric class or the mode for a nominal class)[32] | Baseline | 1 |

For feature selection, we used 14 algorithms that had been implemented in ShinyLearner [89]. Table 1 lists each algorithm, along with a description and high-level category for each algorithm. S12 Data lists hyperparameters evaluated for these algorithms.

For all software implementations that supported it, we set the hyperparameters so that the classification algorithms would produce probabilistic predictions and use a single process/thread. Unless otherwise noted, we used default hyperparameter values for each algorithm, as dictated by the respective software implementations. For feature selection, we used *n_features_to_select = 5* and *step = 0.1* for the `sklearn/random_forest_rfe` and `sklearn/svm_rfe` methods to balance computational efficiency with the size of the datasets. For `sklearn/random_forest_rfe`, we specified *n_estimators = 50* because execution failed when fewer estimators were used.

To analyze the benchmark results, we wrote scripts for Python (version 3.6)[126] and the R statistical software (version 4.02)[127]. We also used the corrplot[128], cowplot[129], ggrepel [130], and tidyverse[131] packages.

## Analysis phases

We performed this study in five phases (Fig 1). In each phase, we modulated either the data used or the optimization approach. In Analysis 1, we used gene-expression predictors only and default hyperparameter values for each classification algorithm. In Analysis 2, we used clinical predictors only and default hyperparameter values for each classification algorithm. In Analysis 3, we used gene-expression and clinical predictors and default hyperparameter values. In Analysis 4, we used both types of predictors and selected hyperparameter values via nested cross-validation. In Analysis 5, we used both types of predictors and selected the most relevant *n* features via nested cross validation before performing classification.

In each phase, we used Monte Carlo cross validation. For each iteration, we randomly assigned the patient samples to either a training set or test set, stratified by class. We assigned approximately 2/3 of the patient samples to the training set. We then made predictions for the test set and evaluated the predictions using diverse metrics (see below). We repeated this process (an iteration) multiple times and used the iteration number as a random seed when assigning samples to the training or test set (unless otherwise noted). ShinyLearner relays this seed to the underlying algorithms, where applicable.

During Analysis 1, we evaluated the number of Monte Carlo iterations that would be necessary to provide a stable performance estimate. For the `mlr/randomForest`, `sklearn/svm`, and `weka/Bagging` classification algorithms, we executed 100 iterations for datasets GSE10320 (predicting relapse vs. non-relapse for Wilms tumor patients) and GSE46691

(predicting early metastasis following radical prostatectomy). As the number of iterations increased, we calculated the cumulative average of the AUROC for each algorithm. After performing at most 40 iterations, the cumulative averages did not change more than 0.01 over sequences of 10 iterations (S32 and S33 Figs). To be conservative, we used 50 iterations in Analysis 1, Analysis 2, and Analysis 3. In Analysis 4 and Analysis 5, we used 5 iterations because hyperparameter optimization and feature selection are CPU and memory intensive. When optimizing hyperparameters (Analysis 4), we used Monte Carlo cross validation for each training set (5 nested iterations) to estimate which hyperparameter combination was most effective for each classification algorithm; we used AUROC as a metric in these evaluations. When performing feature selection (Analysis 5), we used nested Monte Carlo cross validation (5 iterations). In each iteration, we ranked the features using each feature-selection algorithm and performed classification using the top-$n$ features. We repeated this process for each classification algorithm and used $n$ values of 1, 10, 100, 1000, and 10000. For a given combination of feature-selection algorithm and classification algorithm, we identified the $n$ value that resulted in the highest AUROC. We used this $n$ value in the respective outer fold. Finally, when identifying the most informative features across Monte Carlo iterations, we used the Borda Count method to combine the ranks[74].

While executing each analysis phase, we encountered some situations in which we obtained no valid results for all combinations of class variable and algorithms, as noted below.

*Analysis 1*. On iteration 34, the *weka/RBFNetwork* algorithm did not converge after 24 hours of execution time for one of the datasets. We manually changed the random seed from 34 to 134, and it converged in minutes.

*Analysis 2*. The *mlr/glmnet* algorithm failed three times due to an internal error. We limited the results for this algorithm to the iterations that completed successfully.

*Analysis 3*. On iteration 34, the *weka/RBFNetwork* algorithm did not converge after 24 hours of execution time for one of the datasets. We manually changed the random seed from 34 to 134, and it converged in minutes.

*Analysis 4*. During nested Monte Carlo cross validation, we specified a time limit of 168 hours under the assumption that some hyperparameter combinations would be especially time intensive. A total of 1022 classification tasks failed either due to this limit or due to small sample sizes. We ignored these hyperparameter combinations when determining the top-performing combinations. Most failures were associated with the mlr/h2o.gbm and mlr/ksvm classification algorithms.

*Analysis 5*. During nested Monte Carlo cross validation, we specified a time limit of 168 hours. A total of 1408 classification tasks failed either due to this limit or due to small sample sizes. We ignored these tasks when performing hyperparameter optimization.

## Computing resources

We performed these analyses using Linux servers supported by Brigham Young University's Office of Research Computing and Life Sciences Information Technology. In addition, we used virtual servers in Google's Compute Engine environment supported by the Institute for Systems Biology and the United States National Cancer Institute Cancer Research Data Commons[132]. When multiple central-processing cores were available on a given server, we executed tasks in parallel using GNU Parallel [133].

## Performance metrics

In outer cross-validation folds, we used diverse metrics to quantify classification performance. These included accuracy (proportion of accurate predictions), AUROC[134], AUPRC,

balanced accuracy (proportion of accurate predictions weighted by class-label frequency), Brier score[135], F1 score[136], false discovery rate (false positives divided by total number of positives), false positive rate, Matthews correlation coefficient[137], mean misclassification error (MMCE), negative predictive value, positive predictive value (precision), and recall (sensitivity). Many of these metrics require discretized predictions; we relied on the machine-learning packages that implemented each algorithm to convert probabilistic predictions to discretized predictions.

## Supporting information

**S1 Fig. Relative performance of classification algorithms using gene-expression predictors and area under the receiver operating characteristic curve as the metric.** We predicted patient states using gene-expression predictors only (Analysis 1). For each combination of dataset, class variable, and classification algorithm, we calculated the arithmetic mean of area under the receiver operating characteristic curve (AUROC) values across 50 iterations of Monte Carlo cross-validation. Next, we sorted the algorithms based on the average rank across all dataset/class combinations. Each data point that overlays the box plots represents a particular dataset/class combination. The top 15 performers (relatively low ranks) were algorithms that use linear decision boundaries, kernel functions, and/or ensembles of decision trees.
(PDF)

**S2 Fig. Relative performance of classification algorithms using gene-expression predictors and classification accuracy as the metric.** We predicted patient states using gene-expression predictors only (Analysis 1). For each combination of dataset, class variable, and classification algorithm, we calculated the arithmetic mean of classification accuracy across 50 iterations of Monte Carlo cross-validation. Next, we sorted the algorithms based on the average rank across all dataset/class combinations. Each data point that overlays the box plots represents a particular dataset/class combination.
(PDF)

**S3 Fig. Relative performance of classification algorithms using gene-expression predictors and Matthews Correlation Coefficient as the metric.** We predicted patient states using gene-expression predictors only (Analysis 1). For each combination of dataset, class variable, and classification algorithm, we calculated the arithmetic mean of the Matthews Correlation Coefficient across 50 iterations of Monte Carlo cross-validation. Next, we sorted the algorithms based on the average rank across all dataset/class combinations. Each data point that overlays the box plots represents a particular dataset/class combination.
(PDF)

**S4 Fig. Relative performance of classification algorithms using gene-expression predictors and area under the precision-recall curve as the metric.** We predicted patient states using gene-expression predictors only (Analysis 1). For each combination of dataset, class variable, and classification algorithm, we calculated the arithmetic mean of area under the precision-recall curve across 50 iterations of Monte Carlo cross-validation. Next, we sorted the algorithms based on the average rank across all dataset/class combinations. Each data point that overlays the box plots represents a particular dataset/class combination.
(PDF)

**S5 Fig. Comparison of area under the receiver operating characteristic curve (AUROC) and area under the precision-recall curve (AUPRC) scores for Analysis 1.**
(PDF)

**S6 Fig. Comparison of area under the receiver operating characteristic curve (AUROC) and area under the precision-recall curve (AUPRC) scores for Analysis 1, based on ranks (relative performance per algorithm).**
(PDF)

**S7 Fig. Pairwise correlations of sample-level, probabilistic predictions between classification algorithms for dataset GSE10320.** We used each classification algorithm to make probabilistic predictions of relapse in Wilms tumor patients (GSE10320). Based on these predictions, we calculated the Spearman correlation coefficient for each pair of algorithms. These coefficients, averaged across Monte Carlo cross-validation iterations, are illustrated as a correlation plot, clustered based on similarity.
(PDF)

**S8 Fig. Pairwise correlations of sample-level, probabilistic predictions between classification algorithms for dataset GSE46691.** We used each classification algorithm to make probabilistic predictions of early metastasis following radical prostatectomy (GSE46691). Based on these predictions, we calculated the Spearman correlation coefficient for each pair of algorithms. These coefficients, averaged across Monte Carlo cross-validation iterations, are illustrated as a correlation plot, clustered based on similarity.
(PDF)

**S9 Fig. Dataset performance by class category when using gene-expression predictors.** For each class variable across all datasets, we assigned a category representing the type of patient state being predicted. For Analysis 1, we show the predictive performance for each combination of dataset, class variable, and classification algorithm in each class category. We use area under the receiver operating characteristic curve (AUROC) as the metric. The dashed, red line indicates the performance expected by random chance. The top-performing category was "Molecular Marker," which includes class variables associated with mutation status, immunohistochemistry markers of protein expression, presence or absence of chromosomal aberrations, etc. The lowest-performing category was "Patient Characteristic," which includes variables that indicate whether patients had a family history of cancer, had been diagnosed with multiple tumors, patient performance status, etc.
(PDF)

**S10 Fig. Relative performance of classification algorithms using clinical predictors and area under the receiver operating characteristic curve as the metric.** We predicted patient states using clinical predictors only (Analysis 2). For each combination of dataset, class variable, and classification algorithm, we calculated the arithmetic mean of area under the receiver operating characteristic curve (AUROC) values across 50 iterations of Monte Carlo cross-validation. Next, we sorted the algorithms based on the average rank across all dataset/class combinations. Each data point that overlays the box plots represents a particular dataset/class combination (some datasets did not have clinical predictors). The top-performing algorithms (relatively low ranks) were similar overall to Analysis 1; however, some differences were large. For example, weka/NaiveBayes performed best overall in Analysis 2 but was ranked 28th in Analysis 1.
(PDF)

**S11 Fig. Dataset performance by class category when using clinical predictors.** For each class variable across all datasets, we assigned a category representing the type of patient state being predicted. For Analysis 2, we show the predictive performance for each combination of dataset, class variable, and classification algorithm in each class category. We use area under

the receiver operating characteristic curve (AUROC) as the metric. The dashed, red line indicates the performance expected by random chance. The top-performing category was "Diagnosis," which includes class variables associated with a particular disease or subtype. The lowest-performing category was "Patient Characteristic," which includes variables that indicate whether patients had a family history of cancer, had been diagnosed with multiple tumors, patient performance status, etc.
(PDF)

**S12 Fig. Dataset performance by class category when using gene-expression and clinical predictors.** For each class variable across all datasets, we assigned a category representing the type of patient state being predicted. For Analysis 3, we show the predictive performance for each combination of dataset, class variable, and classification algorithm in each class category. We use area under the receiver operating characteristic curve (AUROC) as a metric. The dashed, red line indicates the performance expected by random chance. As with Analysis 1 (S9 Fig), the top-performing category was "Molecular Marker," which includes class variables associated with mutation status, immunohistochemistry markers of protein expression, presence or absence of chromosomal aberrations, etc. The lowest-performing category was "Patient Characteristic," which includes variables that indicate whether patients had a family history of cancer, had been diagnosed with multiple tumors, patient performance status, etc.
(PDF)

**S13 Fig. Relative performance of classification algorithms using gene-expression and clinical predictors.** We predicted patient states using gene-expression and clinical predictors (Analysis 3). For each combination of dataset, class variable, and classification algorithm, we calculated the arithmetic mean of area under the receiver operating characteristic curve (AUROC) values across 50 iterations of Monte Carlo cross-validation. Next, we sorted the algorithms based on the average rank across all dataset/class combinations. Each data point that overlays the box plots represents a particular dataset/class combination.
(PDF)

**S14 Fig. Relative performance of classification algorithms using gene-expression and clinical predictors and performing hyperparameter optimization.** We predicted patient states using gene-expression and clinical predictors with hyperparameter optimization (Analysis 4). We used nested cross validation to estimate which hyperparameter combination would be optimal for each algorithm in each training set. For each combination of dataset, class variable, and classification algorithm, we calculated the arithmetic mean of area under the receiver operating characteristic curve (AUROC) values across 5 iterations of Monte Carlo cross-validation. Next, we sorted the algorithms based on the average rank across all dataset/class combinations. Each data point that overlays the box plots represents a particular dataset/class combination. The algorithm rankings followed similar trends as Analysis 3 (no hyperparameter optimization); however, some differences are notable. For example, the weka/LibLINEAR and mlr/glmnet algorithms were ranked 11th and 16th in Analysis 3 (S13 Fig), but they were ranked 1st and 2nd in this analysis.
(PDF)

**S15 Fig. Dataset performance by class category when using gene-expression and clinical predictors and performing hyperparameter optimization.** For each class variable across all datasets, we assigned a category representing the type of patient state being predicted. For Analysis 4, we show the predictive performance for each combination of dataset, class variable, and classification algorithm in each class category. We use area under the receiver operating characteristic curve (AUROC) as a metric. The dashed, red line indicates the performance

expected by random chance.
(PDF)

**S16 Fig. Correlation between predictive performance and number of samples per dataset.** The number of patient samples differed by dataset. This scatterplot shows the relationship between the median area under the receiver operating characteristic curve (AUROC) and the number of samples in each dataset. We did not observe a significant correlation between these variables.
(PDF)

**S17 Fig. Correlation between predictive performance and number of genes per dataset.** Due to differences in gene-expression profiling platforms, we had data for more genes in some datasets than in others. This scatterplot shows the relationship between the median area under the receiver operating characteristic curve (AUROC) and the number of genes in each dataset. We did not observe a significant correlation between these variables.
(PDF)

**S18 Fig. Variation in predictive performance across hyperparameter combinations.** In Analysis 4, we used nested cross validation to evaluate multiple hyperparameter combinations for each classification algorithm. We assessed the extent to which the area under the receiver operating characteristic curve (AUROC) varied across the hyperparameter combinations for each algorithm. For each combination of dataset, class variable, classification algorithm, and hyperparameter set, we averaged AUROC values across 5 Monte Carlo cross-validation iterations. Then we calculated the coefficient of variation for these averaged values across each combination of dataset/class and classification algorithm. Relatively low values indicate that the hyperparameter sets resulted in similar predictive performance. No results are available for 3 algorithms that used only a single hyperparameter option.
(PDF)

**S19 Fig. Relative performance of different hyperparameter combinations for the weka/ LIBLINEAR classification algorithm.** The ShinyLearner software supports 16 hyperparameter combinations for the weka/LIBLINEAR classification algorithm. In Analysis 4, we used nested cross validation for hyperparameter optimization. For each combination of dataset and class variable, we averaged the area under the receiver operating characteristic curve (AUROC) across all (outer) Monte Carlo cross-validation iterations and then ranked the averages for each hyperparameter combination. Some combinations consistently outperformed other combinations, and the default combination performed suboptimally. Using relatively small cost values appeared to improve the performance more than any other option. This hyperparameter controls the regularization strength.
(PDF)

**S20 Fig. Relative performance of different hyperparameter combinations for the mlr/ glmnet classification algorithm.** The ShinyLearner software supports 3 hyperparameter combinations for the mlr/glmnet classification algorithm. In Analysis 4, we used nested cross validation for hyperparameter optimization. For each combination of dataset and class variable, we averaged the area under the receiver operating characteristic curve (AUROC) across all (outer) Monte Carlo cross-validation iterations and then ranked the averages for each hyperparameter combination. Using an alpha value of 0.5 or 0 resulted in better performance than a value of 1.
(PDF)

**S21 Fig. Relative performance of different hyperparameter combinations for the sklearn/logistic_regression classification algorithm.** The ShinyLearner software supports 32 hyperparameter combinations for the sklearn/logistic_regression classification algorithm. In Analysis 4, we used nested cross validation for hyperparameter optimization. For each combination of dataset and class variable, we averaged the area under the receiver operating characteristic curve (AUROC) across all (outer) Monte Carlo cross-validation iterations and then ranked the averages for each hyperparameter combination. Some combinations consistently outperformed other combinations, and the default combination performed suboptimally. Using relatively small cost values appeared to improve the performance more than any other option. This hyperparameter controls the regularization strength.
(PDF)

**S22 Fig. Relative performance of different hyperparameter combinations for the sklearn/extra_trees classification algorithm.** The ShinyLearner software supports 24 hyperparameter combinations for the sklearn/extra_trees classification algorithm. In Analysis 4, we used nested cross validation for hyperparameter optimization. For each combination of dataset and class variable, we averaged the area under the receiver operating characteristic curve (AUROC) across all (outer) Monte Carlo cross-validation iterations and then ranked the averages for each hyperparameter combination. Some combinations consistently outperformed other combinations, and the default combination performed suboptimally. Using a larger number ($n = 1000$) of estimators (trees) appeared to improve the performance more than any other option.
(PDF)

**S23 Fig. Relative predictive performance when using hyperparameter optimization vs. feature selection.** We used as a baseline the predictive performance that we attained using default hyperparameters for the classification algorithms (Analysis 3). We quantified predictive performance using the area under the receiver operating characteristic curve (AUROC). This graph shows the increase or decrease in performance when selecting hyperparameters or selecting features relative to the baseline. Each point represents a particular combination of dataset and class variable. Generally, the dataset/class combinations that benefitted from hyperparameter optimization also benefitted from feature selection. However, some dataset/class combinations that did not benefit from hyperparameter optimization *did* benefit from feature selection.
(PDF)

**S24 Fig. Dataset performance by class category when using gene-expression and clinical predictors and performing feature selection.** For each class variable across all datasets, we assigned a category representing the type of patient state being predicted. For Analysis 5, we show the predictive performance for each combination of dataset, class variable, and classification algorithm in each class category. We use area under the receiver operating characteristic curve (AUROC) as a metric. The dashed, red line indicates the performance expected by random chance. The results are similar to those of Analyses 3 and 4 (S12 and S15 Figs).
(PDF)

**S25 Fig. Predictive performance according to the number of features selected via nested cross-validation.** Relative area under the receiver operating character curve (AUROC) values were calculated by comparing against the mean for each combination of classification algorithm and feature-selection algorithm.
(PDF)

**S26 Fig. Relative performance of feature-selection algorithms.** For Analysis 5, we used nested cross validation to estimate which features would be most informative for each algorithm in each training set. For each combination of dataset, class variable, and classification algorithm, we ranked the performance of the feature-selection algorithms based on area under the receiver operating characteristic curve (AUROC) and averaged the rankings across 5 iterations of Monte Carlo cross-validation. Each data point that overlays the box plots represents a particular dataset/class combination. Relatively low average ranks are considered optimal. The weka/Correlation feature-selection algorithm performed best overall.
(PDF)

**S27 Fig. Execution time per feature-selection algorithm.** In Analysis 5, we used nested cross validation to estimate which features were most informative for each training set. We calculated the time (in seconds) required by each feature-selection algorithm to rank the features. Then we averaged these times across all combinations of dataset, class variable, classification algorithm, and (outer) Monte Carlo cross-validation iteration. Some feature-selection algorithms were much more computationally intensive than others.
(PDF)

**S28 Fig. Pairwise correlations of feature ranks between feature-selection algorithms for dataset GSE10320.** We used each feature-selection algorithm to rank the genes based on their informativeness for discriminating between relapse and non-relapse outcomes in Wilms tumor patients (GSE10320). After averaging the ranks across cross-validation iterations, we calculated the Spearman correlation coefficient for the feature ranks produced by each pair of algorithms. These coefficients are illustrated as a correlation plot.
(PDF)

**S29 Fig. Pairwise correlations of feature ranks between feature-selection algorithms for dataset GSE46691.** We used each feature-selection algorithm to rank the genes based on their informativeness for predicting early metastasis following radical prostatectomy (GSE46691). After averaging the ranks across cross-validation iterations, we calculated the Spearman correlation coefficient for the feature ranks produced by each pair of algorithms. These coefficients are illustrated as a correlation plot.
(PDF)

**S30 Fig. Absolute classification performance per combination of feature-selection and classification algorithm.** For each combination of dataset and class variable, we averaged the area under the receiver operating characteristic curve (AUROC) across all Monte Carlo cross-validation iterations. Then for each combination of feature-selection algorithm and classification algorithm, we calculated the median AUROC across all datasets and class variables.
(PDF)

**S31 Fig. Relative performance of classification algorithms using gene-expression and clinical predictors and performing feature selection with hyperparameter optimization.** We predicted patient states using gene-expression and clinical predictors with feature selection and optimization of the feature-selection algorithm hyperparameters (Analysis 6). We used nested cross validation to estimate which features and hyperparameter combinations would be optimal for each algorithm in each training set.
(PDF)

**S32 Fig. Stability of classification performance for increasing numbers of cross-validation iterations on dataset GSE10320.** When using gene-expression predictors (Analysis 1), we estimated the number of Monte Carlo cross-validation iterations that would be sufficient to

characterize algorithm performance. For three classification algorithms, we executed 100 cross-validation iterations on dataset GSE10320 (predicting relapse vs. non-relapse for Wilms tumor patients). As the number of iterations increased, we calculated the cumulative average of the area under the receiver operating characteristic curve (AUROC) for each algorithm. After performing at most 40 iterations, the cumulative averages did not change more than 0.01 over sequences of 10 iterations.
(PDF)

**S33 Fig. Stability of classification performance for increasing numbers of cross-validation iterations on dataset GSE46691.** When using gene-expression predictors (Analysis 1), we estimated the number of Monte Carlo cross-validation iterations that would be sufficient to characterize algorithm performance. For three classification algorithms, we executed 100 cross-validation iterations on dataset GSE46691 (predicting early metastasis following radical prostatectomy). As the number of iterations increased, we calculated the cumulative average of the area under the receiver operating characteristic curve (AUROC) for each algorithm. After performing at most 22 iterations, the cumulative averages did not change more than 0.01 over sequences of 10 iterations.
(PDF)

**S1 Data. Summary of predictive performance per dataset when using gene-expression predictors.** We predicted patient states using gene-expression predictors only (Analysis 1). For each combination of dataset, class variable, and classification algorithm, we calculated the arithmetic mean of area under the receiver operating characteristic curve (AUROC) values across 50 iterations of Monte Carlo cross-validation. Next, we calculated the minimum, first quartile (Q1), median, third quartile (Q3), and maximum for these values across the algorithms. Finally, we sorted the algorithms in descending order based on median values. Each row represents a particular dataset/class combination.
(XLSX)

**S2 Data. Summary of predictive performance per dataset when using clinical predictors.** We predicted patient states using clinical predictors only (Analysis 2). For each combination of dataset, class variable, and classification algorithm, we calculated the arithmetic mean of area under the receiver operating characteristic curve (AUROC) values across 50 iterations of Monte Carlo cross-validation. Next, we calculated the minimum, first quartile (Q1), median, third quartile (Q3), and maximum for these values across the algorithms. Finally, we sorted the algorithms in descending order based on median values. Each row represents a particular dataset/class combination. For some dataset/class combinations, no clinical predictors were available; these combinations are excluded from this file.
(XLSX)

**S3 Data. Summary of predictive performance per dataset when using gene-expression and clinical predictors.** We predicted patient states using gene-expression and clinical predictors (Analysis 3). For each combination of dataset, class variable, and classification algorithm, we calculated the arithmetic mean of area under the receiver operating characteristic curve (AUROC) values across 50 iterations of Monte Carlo cross-validation. Next, we calculated the minimum, first quartile (Q1), median, third quartile (Q3), and maximum for these values across the algorithms. Finally, we sorted the algorithms in descending order based on median values. Each row represents a particular dataset/class combination. For some dataset/class combinations, no clinical predictors were available; these combinations are excluded from this file.
(XLSX)

**S4 Data. Summary of predictive performance per dataset when using gene-expression and clinical predictors and performing hyperparameter optimization.** We predicted patient states using gene-expression and clinical predictors (Analysis 4). For classification algorithms that included multiple hyperparameter combinations (n = 47), we performed hyperparameter optimization using the respective training sets. For each combination of dataset, class variable, and classification algorithm, we calculated the arithmetic mean of area under the receiver operating characteristic curve (AUROC) values across 5 (outer) iterations of Monte Carlo cross-validation. Next, we calculated the minimum, first quartile (Q1), median, third quartile (Q3), and maximum for these values across the algorithms. Finally, we sorted the algorithms in descending order based on median values. Each row represents a particular dataset/class combination.
(XLSX)

**S5 Data. Summary of predictive performance per dataset when using gene-expression and clinical predictors and performing feature selection.** We predicted patient states using gene-expression and clinical predictors (Analysis 5). Using each respective training set, we performed feature selection for each of 14 feature-selection algorithms and performed classification using *n* top-ranked features. For each combination of dataset, class variable, and classification algorithm, we calculated the arithmetic mean of area under the receiver operating characteristic curve (AUROC) values across 5 (outer) iterations of Monte Carlo cross-validation. Next, we calculated the minimum, first quartile (Q1), median, third quartile (Q3), and maximum for these values across the algorithms. Finally, we sorted the algorithms in descending order based on median values. Each row represents a particular dataset/class combination.
(XLSX)

**S6 Data. Summary of predictive performance per dataset when using gene-expression and clinical predictors and performing feature selection with hyperparameter optimization.**
(XLSX)

**S7 Data. Top 50 genes according to average rank across feature-selection algorithms for GSE10320 and GSE46691.**
(XLSX)

**S8 Data. Gene-set overlap results for top 50 genes according to average rank across feature-selection algorithms for GSE10320.**
(XLSX)

**S9 Data. Gene-set overlap results for top 50 genes according to average rank across feature-selection algorithms for GSE46691.**
(XLSX)

**S10 Data. Summary of datasets used.** This file contains a unique identifier for each dataset, indicates whether gene-expression microarrays or RNA-Sequencing were used to generate the data, and indicates the name of the class variable from the original dataset. In addition, we assigned standardized names and categories as a way to support consistency across datasets. The file lists any clinical predictors that were used in the analyses as well as the number of samples and genes per dataset.
(XLSX)

**S11 Data. Classification algorithm hyperparameter combinations.** This file indicates all hyperparameter combinations that we evaluated via nested cross-validation in Analysis 4.
(XLSX)

**S12 Data. Feature-selection algorithm hyperparameter combinations.** This file indicates all hyperparameter combinations that we evaluated via nested cross-validation in the follow-up analysis to Analysis 5.
(XLSX)

## Acknowledgments

Results from this study are in part based upon data generated by TCGA and managed by the United States National Cancer Institute and National Human Genome Research Institute (see http://cancergenome.nih.gov). We thank the patients who participated in this study and shared their data publicly. We thank the Fulton Supercomputing Laboratory at Brigham Young University for providing computational facilities. This work was supported in part with a cloud credits allocation provided by the ISB-CGC Cloud Resource, part of the NCI Cancer Research Data Commons.

## Author Contributions

**Conceptualization:** Stephen R. Piccolo.

**Data curation:** Stephen R. Piccolo, Avery Mecham, Nathan P. Golightly.

**Formal analysis:** Stephen R. Piccolo, Nathan P. Golightly.

**Investigation:** Stephen R. Piccolo.

**Methodology:** Stephen R. Piccolo, Nathan P. Golightly.

**Project administration:** Stephen R. Piccolo.

**Resources:** Stephen R. Piccolo.

**Software:** Stephen R. Piccolo, Nathan P. Golightly.

**Supervision:** Stephen R. Piccolo.

**Visualization:** Stephen R. Piccolo, Avery Mecham, Dustin B. Miller.

**Writing – original draft:** Stephen R. Piccolo, Jérémie L. Johnson.

**Writing – review & editing:** Stephen R. Piccolo, Avery Mecham, Nathan P. Golightly, Jérémie L. Johnson, Dustin B. Miller.

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
