## [Decision Letter · Decision Letter 0]

23 Jul 2021

Dear Dr. Piccolo,

Thank you very much for submitting your manuscript "Benchmarking 50 classification algorithms on 50 gene-expression datasets" for consideration at PLOS Computational Biology.

As with all papers reviewed by the journal, your manuscript was reviewed by members of the editorial board and by several independent reviewers. In light of the reviews (below this email), we would like to invite the resubmission of a significantly-revised version that takes into account the reviewers' comments.

We cannot make any decision about publication until we have seen the revised manuscript and your response to the reviewers' comments. Your revised manuscript is also likely to be sent to reviewers for further evaluation.

Sincerely,

Xing Chen, Ph.D.

Guest Editor

PLOS Computational Biology

Edwin Wang

Benchmarking Editor

PLOS Computational Biology

Reviewer's Responses to Questions

**Comments to the Authors:**

Reviewer #1: Line 344: Given the trend you have observed on the datasets you have used using clinical and/or gene expression data, please provide examples, if possible, of other algorithms that have not been studied but could potentially be promising, and why.

Line 356: Trying out convolutional neural networks in deep learning with optimizing number of layers and a hyperparameter search would be useful.

Line 370: The remark about class imbalance being handled well by sklearn is interesting and valid.

Line 377: It is interesting that data on co-occuring tumors did not have significance in feature selection.

Reviewer #2: Piccolo et al. benchmark 50 common classification algorithms from multiple publicly available packages to evaluate algorithm performance in a robust comparative framework. This is a very nice study. While much of their results are not particularly surprising (e.g., parameter optimization improves performance), I believe this study will be an important resource for a broad research community and one that is appropriate for publication in PLOS Computational Biology. I have a few suggestions that I feel would improve the utility and breadth of audience for this work:

1) The AUROC analysis of this manuscript is great and will be a beneficial set of benchmarks for many studies. As the authors acknowledge, however, many studies have unbalanced classes that may see poor results compared to those expected from considering only auROC scores. To address this (and make their results more broadly applicable), it would be nice to see precision-recall curves in addition to their current analyses. The authors should have the data already to generate these plots, so I believe this should be a relatively easy addition.

2) While I appreciate the author’s focus on classification algorithms for classifying biomedical datasets, I believe there could be more attention given to other uses for classification algorithms. A discussion of classification algorithms as a discovery tool—such as using feature selection to identify potentially novel disease or phenotype-associated genes—would increase the breadth of their audience. Since the quality of feature selection is always dependent on the quality of the classification algorithm, but feature extraction is not equally accessible for all algorithms, this could lead to a very interesting additional contrast for the algorithms studied. The authors do touch on feature selection a bit, but mostly in reference to classification. It could be useful to have a brief discussion of how these algorithms perform for feature selection in a discovery context.

3) I like that the authors contrast algorithm performance with running time. That said, I’m less certain that execution time should be valued as strongly as performance. Unless the difference is a matter of days or weeks, I suspect that pretty much all studies would choose the highest quality predictions over a modestly shorter runtime. Outside of (possibly) a few real-world clinical scenarios, I suspect the vast majority of studies would choose high quality predictions over even substantially longer runtimes.

Minor comment: Figure 3 claims the y-axis is log10 transformed, but this does not seem to match the values along the axis.

Reviewer #3: however, the great efforts of the author, the research is poorly organized. it's hard to get benefits for naive researcher?

2. is svm-rfe multivariate? kindly check the type of feature selection methodss

Reviewer #4: In this paper, the authors performed a benchmark comparison, applying 50 classification algorithms to 50 gene-expression datasets (143 class variables). The findings illustrate that algorithm performance varies considerably when other factors are held constant and thus that algorithm selection is a critical step in biomarker studies. The review paper may useful for the researchers and students who interested especially in the fields of characteristic genes of tumor. However, a minor revision is required as indicated below:

1. The selection of tumor characteristic genes is a NP problem. Generally, feature selection algorithms can be divided into three categories: filter, wrapper and embedded. The wrapper method has the advantages of large search space coverage, more flexible classification accuracy and computational efficiency. Wrapper method uses meta heuristic algorithm to obtain the optimal feature subset, and combines the classification algorithm of machine learning as the evaluation standard, which achieves good results in feature selection of high-dimensional medical and health data. The authors should add the analysis of tumor gene feature selection using the meta heuristics.

2. It is suggested that the authors should simplify the introduction and make a more detailed analysis of the discussion.

3. Traditional machine learning methods need to adjust super parameters in feature selection, so it is difficult to determine the best combination of parameters by the analytic method. So that the setting of optimal parameters itself is an optimization problem. Therefore, the parameter setting of the algorithm is worth exploring, and the author should give a detailed discussion.

4. It is suggested that the authors should provide the source programs of all 50 algorithms for better understanding and application of these methods.

**Have the authors made all data and (if applicable) computational code underlying the findings in their manuscript fully available?**

Reviewer #1: Yes

Reviewer #2: Yes

Reviewer #3: None

Reviewer #4: Yes

PLOS authors have the option to publish the peer review history of their article (what does this mean?). If published, this will include your full peer review and any attached files.

Reviewer #1: No

Reviewer #2: No

Reviewer #3: **Yes: **muhammed abd-elnaby sadek

Reviewer #4: No
---

## [Decision Letter · Decision Letter 1]

15 Feb 2022

Dear Dr. Piccolo,

We are pleased to inform you that your manuscript 'The ability to classify patients based on gene-expression data varies considerably across algorithms and performance metrics' has been provisionally accepted for publication in PLOS Computational Biology.

Best regards,

Xing Chen, Ph.D.

Guest Editor

PLOS Computational Biology

Edwin Wang

Benchmarking Editor

PLOS Computational Biology

Reviewer's Responses to Questions

**Comments to the Authors:**

Reviewer #1: The revision made by the authors of the study, with respect to the analysis and additional results, has been satisfactory and serves the scope of the study well. It is noteworthy that added the two neural-network based classification algorithms in their analysis. Yes, the approach to use CNNs to classify gene expession data is emerging. Overall, I think the research in the manuscript is well-organized, carefully done and useful for a broader audience than before. I appreciate the discussion on gene discovery in addition to benchmarking as well. I recognize the useful work done towards the revision in various sections of the manuscript.

Reviewer #2: The authors have done an excellent job of addressing my concerns and those of the other reviewers in my opinion.

Reviewer #4: The paper has been revised according to the revision suggestions, and we have no comments.

**Have the authors made all data and (if applicable) computational code underlying the findings in their manuscript fully available?**

Reviewer #1: Yes

Reviewer #2: Yes

Reviewer #4: None

PLOS authors have the option to publish the peer review history of their article (what does this mean?). If published, this will include your full peer review and any attached files.

Reviewer #1: No

Reviewer #2: No

Reviewer #4: No

---

## [Editor Report · Acceptance letter]

7 Mar 2022

PCOMPBIOL-D-21-00860R1 

The ability to classify patients based on gene-expression data varies by algorithm and performance metric

Dear Dr Piccolo,

I am pleased to inform you that your manuscript has been formally accepted for publication in PLOS Computational Biology. Your manuscript is now with our production department and you will be notified of the publication date in due course.

With kind regards,

Katalin Szabo
